# Hemorrhagic Transformation of Ischemic Strokes

**DOI:** 10.3390/ijms241814067

**Published:** 2023-09-14

**Authors:** Kitti Bernadett Kovács, Viktor Bencs, Lilla Hudák, László Oláh, László Csiba

**Affiliations:** Department of Neurology, Faculty of Medicine, University of Debrecen, 4032 Debrecen, Hungary; kovacs.kitti@med.unideb.hu (K.B.K.); bencs.viktor@med.unideb.hu (V.B.); hudak.lilla@med.unideb.hu (L.H.); olah@med.unideb.hu (L.O.)

**Keywords:** ischemic stroke, hemorrhagic transformation, clinical risk factors, pathophysiology, biomarkers, clinicopathological studies, antithrombotic treatment

## Abstract

Ischemic stroke, resulting from insufficient blood supply to the brain, is among the leading causes of death and disability worldwide. A potentially severe complication of the disease itself or its treatment aiming to restore optimal blood flow is hemorrhagic transformation (HT) increasing morbidity and mortality. Detailed summaries can be found in the literature on the pathophysiological background of hemorrhagic transformation, the potential clinical risk factors increasing its chance, and the different biomarkers expected to help in its prediction and clinical outcome. Clinicopathological studies also contribute to the improvement in our knowledge of hemorrhagic transformation. We summarized the clinical risk factors of the hemorrhagic transformation of ischemic strokes in terms of risk reduction and collected the most promising biomarkers in the field. Also, auxiliary treatment options in reperfusion therapies have been reviewed and collected. We highlighted that the optimal timing of revascularization treatment for carefully selected patients and the individualized management of underlying diseases and comorbidities are pivotal. Another important conclusion is that a more intense clinical follow-up including serial cranial CTs for selected patients can be recommended, as clinicopathological investigations have shown HT to be much more common than clinically suspected.

## 1. Introduction

Ischemic stroke (IS) is usually caused by arterial occlusion in the central nervous system. It is among the leading causes of death and disability worldwide [1]. Hemorrhagic transformation (HT), that is, extravasation of blood into the ischemic tissue, is a serious complication worsening outcomes and increasing mortality. HT can develop by the natural course of IS or after reperfusion therapy [2]. HT occurs at different rates, varying between 3 and 40% depending on the definition used in different studies [3]. According to the European Cooperative Acute Stroke Study (ECASS), HT on CT scans is divided into two stages: radiologically detectable petechiae can be referred to as hemorrhagic infarction (HI), while more severe forms occur as parenchymal hematoma (PH), with or without mass effect [4]. The National Institutes of Neurological Disorders and Stroke (NINDS) classified HT into asymptomatic and symptomatic forms [5]. The SITS-MOST criteria define symptomatic intracranial hemorrhage (sICH) as a local or remote type 2 parenchymal hematoma occurring within 22 to 36 h post-thrombolysis on a CT scan and is associated with an increase in NIHSS of four points from the baseline or resulting death [6]. Based on the pathophysiology, we aimed to collect the main factors contributing to hemorrhagic transformation, including clinical risk factors, laboratory parameters, and biomarkers. We also searched the promising experimental data on possible treatment options for the future.

## 2. Pathophysiology of HT

The integrity of the blood–brain barrier (BBB), i.e., the physiological barrier between the circulating blood and brain interstitium, is crucial. This layer is composed of several cells, including endothelial cells, astrocytes, pericytes, neurons, and extracellular matrix (ECM), forming the so-called neurovascular unit (NVU) [7]. In addition to these cells, basement membranes with tight, adherens, and gap junctions between the cells are also important components of the BBB, which serve as a bidirectional barrier in the transport of different substances and protect brain parenchyma from harmful chemicals. Tight junctions limit the paracellular movement of water, ions, and solutes, and form a barrier to substances with a molecular weight greater than 180 Da. The endothelial cells serve as the first-line defense between the circulating blood and brain parenchyma. They are crucial in the regulation of ion movement and the maintenance of selective molecular permeability and integrity. For this role, endothelial cells have an increased number of mitochondria allowing them to generate the high amount of energy required for the maintenance of proper nutrient support and protection of the brain. Therefore, endothelial cells are very sensitive to an optimal oxygen and glucose supply. BBB endothelial cells are distinguished from peripheral ones by their lack of fenestrations and minimal pinocytotic activity permitting much less transcellular (caveolar) transport compared to the peripheral circulation [7,8]. The interaction of pericytes with endothelial cells is essential for the stability of the BBB. These cells are surrounded by the ECM, providing support and separation, and regulating intercellular communication. The ECM is composed of structural proteins, like laminin, fibronectin, collagen type-IV, elastin, trombospondin, and proteoglycanes, which are susceptible to enzymatic degradation. Astrocytes are also important components of the BBB. Their endfeet are rich in aquaporin-4 water channels giving them an important role in the regulation of brain water content and electrolyte balance. Early ischemia, via ruining these structures and processes, disrupts the BBB [7,9]. Components of neuroinflammatory mechanisms are activated involving microglia and astrocyte activation and the secretion of proinflammatory cytokines and growth factors like TNF-α, IL-1β, IL-17, INFγ, VEGF, and matrix metalloproteinases (MMPs) [10]. Neutrophils are recruited to the damaged area; they adhere to the vascular endothelial cells, migrate into the brain parenchyma, and their activation adds to the leakage of the BBB via several ways including the production of matrix metalloproteinase-9 (MMP-9), an enzyme participating in the degradation of the BBB [11,12]. Lymphocytes infiltrate the brain after neutrophils, releasing different pro-and anti-inflammatory cytokines to regulate the BBB [11]. Monocytes, after transferring into the central nervous system, differentiate into macrophages, promoting the expression of certain factors protecting the BBB and contributing to healing mechanisms [13]. A schematic representation of the NVU and BBB and their disruption due to ischemic damage is shown in Figure 1. Along with the loss of BBB function and the consequent edema formation, the autoregulatory capacity of cerebral vasculature is also weakened. This ability of the cerebral vessels is responsible for the preservation of stable blood flow despite wide variations in systemic blood pressure [14]. Both pericytes, due to their contractile properties, and astrocytes, via their endfeet contacts, contribute to the regulation of capillary blood flow and the maintenance of cerebrovascular autoregulation [7]. In acute stroke, the autoregulatory capacity is of pivotal importance, as it ensures perfusion to the ischemic penumbra and also helps to avoid or moderate reperfusion injury that may happen spontaneously or after thrombolytic and endovascular treatment [14,15,16]. Autoregulatory impairment occurs even in case of minor stroke and may exist ipsilaterally to the affected side but can also be a global phenomenon in both hemispheres [15].

Following ischemia, another pathological impact of ischemic stroke is reperfusion occurring spontaneously or after reperfusion therapies. Nevertheless, reperfusion is required for tissue survival; it may also bring on further tissue damage with the potential of HT. Based on experimental data, three phases of restored blood flow may occur, which are presented in Table 1. The initial reperfusional permeability is attributable to an acute elevation of regional cerebral blood flow. This hyperemia, contributing to the acute opening of the endothelial tight junctions, is followed by a period of hypoperfusion with a biphasic permeability response. The hypoperfusion can be attributed to metabolic depletion, microvascular obstruction by cumulating leukocytes, endothelial swelling, and disintegration of the NVU. Hypoperfusion of the ischemic area further results in deficient nutritional support. Enhanced neutrophil adhesion and subsequent inflammatory mechanisms lead to the next period of increased BBB permeability (the first phase of the biphasic permeability 3–6 h after reperfusion). Angiogenesis and increased vasogenic edema are components of the second phase of the biphasic BBB permeability 18–96 h after reperfusion. The duration of the ischemia, degree of reperfusion, and the applied stroke model influence the multiphasic nature of this process. In the ischemic phase, the disturbance of energy production of the cells leads to several pathological processes including failure of the sodium potassium pump and the accumulation of sodium and calcium in the cells causing intracellular translocation of interstitial water resulting in cytotoxic edema. Alternative, less efficient ways of energy production are activated resulting in lactacidosis, directly contributing to the swelling of the cells of the NVU. As a neuronal component, excitotoxic glutamate is released from neurons. Endothelial swelling leads to a shrinking in capillary diameter, further deteriorating blood supply. The induction of proteases like endogenous tissue plasminogen activator and MMPs along with the release of proinflammatory cytokines, chemokines, and adhesion molecules initiate leukocyte infiltration and inflammatory activation. During reperfusion, vasogenic edema develops due to alterations in BBB tight junctions, as an increasing permeability of macromolecules allows fluid movement from intravascular to extravascular spaces. Tight junctions begin to go through a regulated period of disassembly and reassembly between existing endothelial cells, while new assembly is related to new cell growth as angiogenesis starts in the final, biphasic BBB permeability. Vasogenic edema increases overall brain volume with a peak 2–5 days after ischemic stroke; however, neurovascular remodeling may continue weeks after the initial ischemic event [7,9,17].

As summarized above, the pathophysiological basis of HT is the disintegration of the BBB and NVUs. Increased BBB leakage promotes extravasation of blood cells initiated by platelets recruiting leukocytes to the sites of their extravasation. The flow of leukocytes decelerates, they start rolling on the activated endothelium, and finally paracellular transmigration happens. This process has been shown to be increased by the acute phase protein fibrinogen in a dose-dependent manner. If the vascular wall is damaged significantly enough, erythrocytes also extravasate after BBB breakdown, resulting in the deposition of hemoglobin-derived neurotoxic products like free iron in the brain parenchyma. The reactive ferrous form of iron triggers lipid peroxidation and oxidative damage resulting in neuronal death and brain edema. Iron-derived reactive oxygen species further alter BBB permeability and promote further transmigration of blood cells. Fibrinogen, apart from its promoting effect on leukocyte extravasation, increases the aggregation of erythrocytes. The source of fibrinogen is the circulation, as it is mainly synthesized by the liver and stored in both the alpha and dense granules of platelets. Upon platelet activation, the deposition of the released fibrinogen serves as a binding site for yet non-activated platelets adding to platelet thrombogenesis and erythrocyte aggregation leading to reduced blood flow and also diminution of the hematoma developing upon HT [7]. Depending on the severity of the extravasation of blood cells, petechial and parenchymal hemorrhages can be distinguished. Petechial hemorrhage reflects small (HI1) or more confluent (HI2) pinpoint bleeds within the ischemic region. Parenchymal hemorrhage is a more serious form of bleeding involving a larger part of the infarcted area (≤30% in the case of PH1 with a mild space-occupying effect, >30% in the case of PH2 with a significant space-occupying effect) [4].

In the subsequent sections of this article, we will review the clinical risk factors of HT based on the aforementioned pathological processes, with an emphasis on their therapeutic options.

## 3. Clinical Risk Factors of HT

Hypertension, early ischemic signs on CTs, poor collateral circulation, reperfusion therapy, hyperglycemia, severe forms of ischemic stroke (higher NIHSS points), advanced age, low platelet count, and antithrombotic treatment have been known to increase the risk of HT after acute ischemic stroke [17]. Cardioembolic etiology also increases the risk [18]. Table 2 summarizes the clinical risk factors of HT in acute ischemic stroke.

### 3.1. Role of Hypertension

Elevated blood pressure (BP) has been shown to increase the risk of HT via exacerbating inflammation, influencing vascular remodeling, collateral circulation, and autoregulation, and disrupting the BBB partly due to direct pressure exerted on brain vasculature [19]. Current guidelines recommend a blood pressure below 180/105 mmHg for patients undergoing recanalization treatment [20]. However, post-procedural blood pressure control is less well defined. The ESCAPE protocol suggests maintaining systolic BP above 150 mmHg to promote proper collateral flow to penumbra while the artery remains occluded. If reperfusion has been achieved, aiming for normal pressure is reasonable as hyperperfusion could lead to cerebral edema and HT [21]. The DAWN protocol endorses a systolic BP below 140 mmHg in the first 24 h after successful reperfusion [22]. Goyal et al. reported a single-center experience on more aggressive BP control after successful endovascular treatment. Compared to permissive hypertension (systolic BP < 180 mmHg), patients undergoing moderate (<160 mmHg) and intensive (<140 mmHg) BP control showed an improved functional outcome and lower mortality at 90 days [23]. Rather than adhering strictly to specific BP values, clinicians should consider individual patient characteristics, such as the degree of reperfusion, infarct size, presence of carotid artery stenosis, antithrombotic therapy, and hemodynamic status. Personalized management should then be implemented based on these factors [24]. Real-time autoregulation monitoring may be a useful tool to identify a dynamic BP range for individual patients. Wang et al. found that deviation from the limits set by interrogating changes in near-infrared spectroscopy-derived tissue oxygenation (a cerebral blood flow surrogate) in response to changes in mean arterial pressure increased the risk of further brain injury and poor functional outcome. However, the authors emphasize the need of a randomized trial to determine the optimal approach for hemodynamic management [16].

### 3.2. Role of Early Signs of Ischemia, Vessel Calcification, and Hyperdense Middle Cerebral Artery Sign on Cranial CT and Collateral Score on CTA

Sudden ischemia (decreased ATP levels due to failure of oxygen and glucose delivery) causes damage to the BBB resulting in cytotoxic edema and increased permeability in the hyperacute stage (minutes to hours after ischemic stroke), while neuroinflammatory processes further increase the permeability in the acute stage (hours to days after the incident) [9]. Arba et al. concluded the Reperfusion Injury in Ischemic Stroke (RISK) study aimed at assessing BBB leakage within the ischemic area using the volume transfer constant value on pretreatment cranial CT images. These values reflect the extravasation (leakage) of contrast material in the ischemic area. They found that BBB disruption was associated with HT and sICH (the magnitude of effect was larger in patients treated later than 4.5 h from symptoms onset) but not with functional outcome at 3 months [25].

CT perfusion imaging has a considerable role in HT prediction, assisting in the selection of patients with a small core and a large penumbra that are less likely to develop sICH [26].

Yu et al. assessed and quantified intracranial calcification of seven main arteries (bilateral internal carotid artery C2–7 segments, middle cerebral arteries, vertebral arteries V4 segments, and basilar artery) on baseline CT scans before rtPA treatment. They found that the calcification volume on the lesion side was associated with HT after thrombolysis, and the higher number of calcified vessels lead to a poorer prognosis [27].

Hyperdense middle cerebral artery sign is one of the most important factors in the SEDAN score to assess the risk of sICH [28]. Zou et al. found it to independently predict HT after thrombolysis [29].

Hao et al., just like the DIRECT-MT trial, showed that poor collateral circulation may increase the risk of HT and sICH [30,31].

### 3.3. Role of Recombinant Tissue Plasminogen Activator (rtPA) and Mechanical Thrombectomy

Recanalization therapies result in a three-phase process of reperfusion detailed above in the pathophysiology section. Restoration of blood flow leads to reactive hyperemia with impairment of cerebral autoregulation again leading to cytotoxic edema. A reactive microvascular obstruction then develops, worsening the function of the BBB. Finally, paracellular permeability increases, associated with vasogenic oedema and angiogenesis [7,9,32]. About 2–6% of the patients undergoing thrombolytic therapy are at risk of developing sICH, while HT remains asymptomatic in 20–40% of those treated [33]. The rtPA has been shown to disrupt BBB integrity via several ways, including LDL receptor-related protein (LRP) expression on the endothelial cells, microglia, and astrocytes [34], increasing plasma kallikrein [35] and platelet-derived growth factor-CC (PDGF-CC) activation [36]. Shi et al. demonstrated that rtPA also mobilizes immune cells exacerbating HT after ischemic stroke [37]. Mechanical thrombectomy may cause direct endothelial denudation, edema in the intima and media, and disruption of the internal elastic lamina [38]. In addition, the intervention may lead to rapid reperfusion with the risk of HT [9]. Five randomized trials (MR CLEAN, ESCAPE, SWIFT PRIME, REVASCAT, and EXTEND IA) examined the efficacy of endovascular treatment (EVT) in the early time window (up to 12 h) and found that sICH in the treatment group ranged between 0 and 7.7%. In these studies, more than 80% of patients received rtPA before EVT [21,39,40,41,42]. In the extended time window trials (up to 24 h), the DAWN and DEFUSE-3, sICH occurred in 6–7% of treated patients [43,44]. The HERMES meta-analysis revealed that sICH is not more frequent in patients receiving EVT than in patients with medical therapy alone [45].

### 3.4. Role of Hyperglycemia

An elevated blood sugar level upon admission has been shown to increase BBB disruption leading to a less favorable outcome and more sICH [46]. The DIRECT-MT trial reported that a high blood glucose level at hospital arrival can be a predictive factor for HT, PH, and sICH in patients undergoing EVT and thrombolysis combined with EVT and is an independent predictor of sICH in the combination therapy group [30]. The SHINE and GIST-UK trials failed to show significant improvement and better outcomes in the treatment of hyperglycemia in the acute setting [47,48]. A potential explanation for this can be read in the cohort of Kerényi et al. concluding that not the acute but the chronic elevation of blood glucose level increases the risk of HT via the more prominent pathological processes around the vessel walls in the BBB [49].

### 3.5. Role of Admission NIHSS Points

Kidwell et al. reported that NIHSS > 15 had a greater than 50% rate of HT after intravenous thrombolysis [50]. Qian et al. identified the admission NIHSS score as a predictive factor for sICH after EVT [51], similar to the DIRECT-MT analysis [30].

### 3.6. Role of Advanced Age

The increased risk in patients with advanced age results from complex issues, like increased systemic inflammation and changes in BBB function and both the innate and adaptive immune system [52,53]. Kerényi et al. found that age is a risk factor of HT for patients whose infarct is of embolic origin. According to their results, the embolic way itself does not necessarily mean a higher risk of HT; a high age—that is 75 and beyond—is also required for this effect to manifest [49]. However, in a single-center study carried out at our department, Héja et al. found that the occurrence of HT, including sICH, among patients undergoing intravenous thrombolysis was similar in patients below and above 80 years, supporting alteplase treatment to be safe for the elderly [54]. To clarify the underlying mechanisms of the latter finding further investigations are required.

### 3.7. Role of Timing

Raychev et al. reported that prolonged time from onset to groin puncture increases the risk of PH and basal ganglia bleeding after EVT within 8 h [55]. Hao et al. showed that if >270 min have passed between symptoms onset and groin puncture, the EVT was associated with an increased risk of sICH [31].

### 3.8. Role of Low Platelet Count and Size

Domingo et al. in the analysis of three retrospective observational studies concluded that the postprocedural clinical outcomes (mRS > 3) after EVT and the risk of the development of sICH were similar in thrombocytopenic patients compared to patients with normal platelet counts. However, the mortality was higher in cases of thrombocytopenia after EVT. Of note, the lack of a clear cutoff value for low platelet count (range of 50,000–100,000/μL in previous studies) represents a challenge during risk stratification. Comorbidities associated with thrombocytopenia (e.g., thrombotic thrombocytopenic purpura, heparin-induced thrombocytopenia, disseminated intravascular coagulation, alcoholic cirrhosis and malignancies) were addressed to be responsible for the increased mortality rate [56]. In a case–control study, Cheng et al. found that low platelet count, low mean platelet volume (MPV), and high fibrinogen (FIB) levels were significant and independent risk factors of HT in non-atrial fibrillation (AF) patients but not in AF patients. They concluded that MPV and FIB levels were independently and significantly associated with unfavorable long-term functional outcomes in non-AF HT patients [57].

### 3.9. Role of Cardioembolism

Hemorrhagic transformation of cerebral infarcts due to cardiogenic embolism seems to be a regular finding in medium-sized and large infarcts even without anticoagulation [58]. Molina et al. concluded that delayed recanalization occurring more than 6 h after acute cardioembolic stroke is an independent predictor of HT [59]. As mentioned previously, along with an advanced age, it is a predictor of HT [49].

### 3.10. Role of Antithrombotic Treatment

Antithrombotic treatment is an essential part of secondary stroke prevention since the coagulation system plays a pivotal role in stroke pathogenesis. Nevertheless, the injured brain tissue and the damaged autoregulation of the cerebral vessels make patients suffering from acute IS prone to have bleeding complications; antiplatelet medications and anticoagulant treatment are warranted for the prevention of further strokes of atherothrombotic and cardioembolic origin, respectively [60]. In general practice, antiplatelet therapy is started 24 h after thrombolysis or EVT as a secondary prevention if the 24 h control CT is free from signs of hemorrhagic complications [61]. In the case of minor atherothrombotic ischemic stroke, in the lack of thrombolytic therapy, dual-antiplatelet therapy with aspirin and clopidogrel is recommended, initiated within 24 h after onset and maintained for 21 days [20]. If stent implantation happens during or independent of EVT, dual-antiplatelet therapy is required to be started immediately, lasting for three months [62]. According to current European Stroke Organization (ESO) guidelines on the pharmacological treatment for the prevention of transient ischemic attack (TIA) and IS, the use of antiplatelet therapy did not significantly increase the risk of hemorrhagic stroke, but the level of certainty was low. This guideline suggests the avoidance of dual-antiplatelet therapy with aspirin and clopidogrel after the first three months [63]. The different antiplatelet medications have their well-established role for the treatment of ischemic heart disease; trials were mainly performed in this field. In studies aimed at investigating the effects of intensified antiplatelet schemes, it was concluded that aspirin in combination with either clopidogrel or ticagrelor decreased the risk of a further ischemic stroke, but the bleeding risk was higher. Of note, no remarkable differences were observed among the applied regiments considering mortality. However, prasugrel was found to significantly increase the rate of intracranial bleeding in acute coronary artery disease patients with previous cerebrovascular events [60].

For patients with concomitant AF, prompt initiation of oral anticoagulation will greatly reduce the risk of a subsequent stroke and other forms of systemic embolization, although caution is needed for the optimal timing considering stroke severity [61]. According to the European Society of Cardiology (ESC) and European Heart Rhythm Association (EHRA) guidelines, which have been endorsed by the ESO, oral anticoagulant therapy should be continued or omitted for one day in the case of TIA, for three days in the case of minor stroke (NIHSS score < 8), for six days in the case of mild stroke (NIHSS score 8–15), and for twelve days in the case of severe stroke (NIHSS score > 15) [64]. In the case of nonvalvular AF, the use of direct thrombin and factor Xa inhibitors, such as apixaban and rivaroxaban, may have less risk of intracranial hemorrhage than patients on warfarin [65]. However, the use of warfarin is indicated in cases of valvular AF, after implantation of biological (for 3 months) and mechanical heart valves (lifelong). The latter case represents such a strong indication of anticoagulation that has to be carried out regardless of the size of the intracranial ischemia or its hemorrhagic transformation [66]. The use of direct oral anticoagulant drugs (DOACs) for an embolic stroke of undetermined origin is not recommended. However, the use of low-dose DOACs in addition to an antiplatelet drug can be considered in patients with concomitant coronary or peripheral artery disease one month after their ischemic stroke or TIA [64].

It is noteworthy to mention that immobilized patients with severe ischemic stroke require a preventive dose of anticoagulants to avoid deep venous thrombosis (DVT) and its potentially fatal consequence, pulmonary embolism (PE). Previous meta-analysis data suggested that it may be most beneficial to treat prophylactically with low-molecular-weight heparin once daily. The risk reduction in DVT and PE comes at the cost of a significant increase in sICH. However, these meta-analysis data suggest that prophylactic doses of anticoagulation did not significantly affect mortality [65].

Powers collected the results of three observational studies that showed no harm in beginning or continuing antithrombotic therapy (both aspirin and acenocoumarol) to patients who have already developed HT; therefore, they concluded that antithrombotic therapy decisions should be independent of the presence or absence of HT [67]. Other researchers came to a similar conclusion, as although intensified platelet regimens increase bleeding rate, mortality remains unaffected [60].

### 3.11. Role of Further Clinical Factors

#### 3.11.1. Liver Fibrosis

Liver fibrosis is associated with an increased risk of hemorrhagic stroke [68] and is a strong predictor of long-term mortality in ischemic stroke [69] partly due to its association with HT, even in the case of normal standard liver enzymes [70]. There might be three underlying mechanisms of the latter: liver fibrosis increases endothelial dysfunction [71], inflammation [72], and oxidative stress [73], all contributing to BBB dysfunction. Subclinical coagulopathy is another contributor to HT [74].

#### 3.11.2. Iron Homeostasis

Iron is essential in the homeostasis of both the whole body and the brain, taking part in such important processes as oxygen transport in the blood, mitochondrial respiration, and neurotransmitter synthesis in the brain [75]. Of note, it can also be toxic via the generation of reactive oxygen species damaging cellular substrates. In order to avoid this, a sequel of homeostatic mechanisms works to keep the iron in the less reactive ferric form and to form a chelate with specific binding proteins. Such processes, along with an intact BBB, keep brain iron levels constant even in the case of disorders causing systemic iron accumulation [76]. However, cerebral ischemia leads to impaired iron homeostasis by releasing iron from its binding proteins to its most reactive ferrous form. Due to the impaired BBB, blood-borne iron may also enter the ischemic parenchyma, worsening the damage [8,77]. Iron overload is a predisposing factor for complications after thrombolytic therapy and exacerbates the risk of HT. Elevated iron burden being relatively common in the population, caused by frequent hemochromatosis, thalassemia, and diabetes mellitus as examples, Yébenes et al. suggests combinations of reperfusion agents and neuroprotective substances as relevant treatment options for a safer thrombolysis, though clinical studies are required [78].

### 3.12. Role of Gender

Won et al. found an association between possible osteoporosis (predicted by the measurement of the Hounsfield unit value of the frontal skull bone on a brain CT) and the development of PH-type HT in patients with cardioembolic stroke, but this was significant in males not females [79]. Recognizing gender-specific risks can have implications for treatment decisions. Tailoring antithrombotic and anticoagulant therapies based on individual risk factors, including gender, may be necessary [79,80].

## 4. Biomarkers of HT

According to the definition of the World Health Organization (WHO), “any substance, structure, or process that can be measured in the body or its products and influence or predict the incidence of outcome or disease” can serve as a biomarker or biological marker [81]. Biomarkers are continuously investigated to help with the prediction of potential adverse effects of stroke treatment like HT, optimizing treatment and patient management. The most promising ones are summarized in Table 3.

### 4.1. Matrix Metalloproteinase-9 (MMP-9)

Matrix MMPs are zinc-binding proteolytic enzymes that take part in remodeling the extracellular matrix (ECM). MMP-9, originating mainly from neutrophils, contribute to BBB dysfunction, attacking type IV collagen, laminin, and fibronectin—the major components of the basal lamina around cerebral blood vessels [2,82]. A meta-analysis concluded that baseline MMP-9 levels were higher in patients who developed sICH and were associated with a poor outcome, so could function as a sensitive and specific marker for the prediction of HT, particularly the severe forms [83]. Based on animal models, MMP inhibition could be potentially used for the reduction (presence and extent) of hemorrhagic complications in connection with rtPA treatment [84,85].

### 4.2. Cellular Fibronectin (c-Fn)

Fibronectin exists in two forms, soluble in the plasma produced by hepatocytes and insoluble or cellular, and is a major component of the ECM, secreted mainly by fibroblasts in the soluble form, which is then assembled into an insoluble matrix [86]. Significant elevation of c-Fn was reported in post-thrombolysis patients with PH by a few studies [12,83].

### 4.3. Ferritin

Baseline ferritin levels were elevated in patients with PH and sICH and were associated with poor outcomes [83]. Being an acute phase protein, its concentration has been shown to increase in response to stresses such as inflammation and hypoxia of various causes [87], calling for the need of determination in more precise settings.

### 4.4. Neutrophil-to-Lymphocyte Ratio (NLR)

Previous studies showed prominent neutrophil infiltration in brain areas affected by both ischemic and hemorrhagic stroke [88]. Lymphocytes accumulate in the infarcted area a few days later than neutrophils. A higher rate of neutrophils can cause a heavier initial stroke, while a lower rate of lymphocytes is associated with a poorer long term prognosis [11], as a relative decrease in lymphocyte number represents a cortisol-induced stress response leading to an increase in proinflammatory cytokines, aggravating ischemic injury [89]. Zhang et al. performed a meta-analysis of the available studies concerning the predictive role of NLR in HT after ischemic stroke. They concluded that a higher NLR cutoff value between 7.5 and 11 had a better predictive value for HT and 90-day mortality and found it a simple, reliable, and promising inflammatory indicator [90]. An other meta-analysis also found a correlation between baseline NLR and poor outcomes after ischemic stroke but concluded that NLR weakly correlated with sICH, indicating a low discriminative ability compared to other markers like MMP-9, c-Fn, and ferritin [83].

### 4.5. S100 Calcium-Binding Protein B (S100B)

This is glial-specific protein expressed primarily by a subtype of mature astrocytes ensheathing blood vessels. Its function mainly lies in astrocytosis, axonal proliferation, and neurite extension, and in adults, its level is usually elevated in the case of central nervous system damage, making it a potential clinical marker [91]. Based on two studies, the baseline levels of S100B were elevated in patients with sICH [83].

### 4.6. Soluble ICAM-1 (sICAM-1)

It is a peptide that comprises the extracellular part of ICAM-1, a transmembrane protein belonging to the immunoglobulin superfamily that is upregulated by proinflammatory cytokines [92]. According to the work of Wu et al., sICAM-1 levels of ischemic stroke patients were significantly higher than healthy controls, and serum sICAM-1 levels of patients with cerebral microbleeds were higher than patients without microbleeds; hence, it might be independently associated with an increased risk of HT in stroke patients [93].

### 4.7. Vascular Adhesion Protein-1 (VAP-1) and Semicarbazide-Sensitive Amine Oxidase (SSAO)

VAP-1 is an adhesion molecule critically involved in the process of leukocyte trafficking to sites of inflammation and possesses SSAO enzyme activity (catalyzing the oxidative deamination of primary amines to produce aldehydes, ammonium, and hydrogen peroxide). Based on animal studies, the endothelium can be a major source of circulating, catalytically active VAP-1/SSAO [94]. Hernandez et al. reported that baseline VAP-1/SSAO activity was associated with the incidence of HT, and treatment with a VAP-1//SSAO inhibitor prevented adverse effects caused by delayed t-PA administration in a rat model [95].

There are several other potential biomarkers mentioned in the literature in single studies, requiring further validation even for research use. Looking over the list above, it is presumable that the measurement of biomarkers is time-requiring except for NLR, which has questionable predictive value concerning the outcome of a stroke and its treatment and currently seems unimaginable to influence therapeutic decisions. Considering the narrow time window for the treatment of ischemic stroke, the baseline determination of biomarkers in the acute clinical setting is barely feasible and is also lacking therapeutic consequence. However, they might help in the patient selection requiring greater attention during hospitalization, including closer follow up and more frequent cranial CT scans to detect HT. Based on the profound meta-analysis of Krishnamoorthy et al., the highest sensitivity was observed for c-Fn and ferritin; the serum levels of both correlated with each other as well as with MMP-9 levels at 24 h from onset in sICH patients [83]. Probably the use of a combination of several biomarkers, instead of single ones, will be the key to increase specificity and predictive value. Further proteomic investigations on plasma, cerebrospinal fluid, and brain tissue samples may also help to find the most suitable markers.

## 5. Role of the Gut Microbiome—Promising Experimental Stroke Models

Increasing clinical and mainly preclinical research using animal models is ongoing in the field of microbiome composition as a possible key factor in the pathophysiology of neurological and psychiatric disorders like Alzheimer’s disease, autism spectrum disorder, multiple sclerosis, Parkinson’s disease, and stroke and post-stroke cognitive impairment [96]. Huang et al. precipitated HT in rats by induced hyperglycemia before a middle cerebral artery occlusion and reperfusion model (actually, by this, they also showed that acute hyperglycemia is associated with increased HT incidence, supporting those mentioned above in the section on the role of hyperglycemia.) The ratio of Proteobacteria and Actinobacteria was higher in the hyperglycemic group, and these rats were enriched with pathogenic and opportunistic pathogens like Erysipelotrichi, Eschericia, Streptomycetaceae, and Pseudonocardiaceae [97]. Other researchers found the proinflammatory Proteobacteria and Actinobacteria to be frequent in stroke patients [98]. In turn, HT increased the ratio of anaerobic bacteria such as Actinobacteria, Proteobacteria, and Synergistetes in the intestinal tract by inhibiting oxygen formation [97]. Lipopolysaccharide (LPS) originating from the microbiota (Proteobacteria, involving Enterobacteriaceae) is translocated into the circulatory system, inducing the release of inflammatory cytokines via Toll-like receptor 4, accelerating systemic inflammation and exacerbating brain infarction [99]. By also the influence of the immune homeostasis, the gut microbiota is a stroke risk factor, potentially increasing the expression of proinflammatory cytokines like IL-17 and IFN-γ [100]. Bidirectional signaling seems to be obvious in the studies and can be referred to as the ”gut–brain axis”, the brain and gut barriers being highly correlated [101,102]. As Huang et al. showed that changes in the microbiota composition were associated with the severity and risk of HT after stroke, cerebral ischemia leads back to an increased intestinal permeability, worsening stroke outcomes [97]. Experiments showing the transplantability of short-chain-fatty-acid-producing bacteria to microbiota-depleted (germ-free or antibiotic-depleted) rodents, thereby improving the outcomes after stroke (particularly butyrate improving neurogenesis and easing brain inflammation), opens up the possibilities of future treatments [103]. The differences of the microbiome spectra of rodents and humans make the interpretation of research data difficult, along with the fact that the composition of the microbiome is influenced by various factors such as diet and exercise [96,97].

## 6. Therapeutic Approaches

Regarding the clinical risk factors, individualized management is reasonable. Blood pressure is one of the key risk factors of HT; therefore, careful monitoring should be carried out, and patient characteristics should be considered during its treatment. Another crucial factor is hyperglycemia, the treatment of which is mandatory, although in acute conditions, it does not improve the outcome. Considering the underlying comorbidities leading to ischemic stroke, the narrow time window given for its treatment, and the complexity of mechanisms promoting HT, the importance of primary prevention needs to be underlined—to reduce all the influenceable risk factors before any ischemic event. As for the inflammatory components, some drugs have been shown to protect the BBB and reduce the risk of HT both in preclinical experiments and clinical trials; however, further evaluation is required for safety and effectivity.

### 6.1. Minocycline

This tetracycline antibody, via the inhibition of microglia activation, decreasing the migration and infiltration of neutrophils to the brain, and suppressing the expression of MMP-9, decreased the risk of HT in animal stroke models [104,105,106].

### 6.2. Fingolimod

It is an immunotherapy for the treatment of relapsing, remitting sclerosis multiplex targeting sphingosine 1-phosphate receptor inhibiting the production of lymphocytes in lymph nodes. Tian et al. found that fingolimod may enhance the efficacy of alteplase treatment in the 4.5 to 6 h time window in patients with a proximal cerebral arterial occlusion, as patients who received fingolimod and alteplase exhibited a greater reduction in the perfusion lesion in a prospective, randomized, open-label, blinded endpoint clinical trial [107]. Fingolimod attenuated HT after cerebral ischemia and reperfusion in a lymphocyte-independent fashion but was ineffective under thrombocytopenic conditions in an animal model [108]. In a pilot study by Zhu et al., a combination therapy of fingolimod and alteplase was well tolerated, attenuated reperfusion injury, and improved clinical outcomes in patients with acute ischemic stroke, although their findings need to be tested in further clinical trials [109].

### 6.3. Tacrolimus

Tacrolimus is an immunosuppressant drug mainly used for the treatment of immunological rejection after transplantation [110]. In an experimental setting, a combined treatment with tPA and tacrolimus vs. tPA alone was conducted in hypertensive rats, leading to the conclusion that the incidence of HT and in the case of HT, hematoma volumes were lower in the combined treatment group [111].

### 6.4. Statins

Atorvastatin given before delayed tPA therapy (up to 6 h) decreased neutrophil infiltration and MMP-9 expression in a rat model [112]. Lu et al. demonstrated that rosuvastatin decreased HT via decreasing BBB leakage, attenuating neuroinflammation (decreased the levels of TNF-α, iNOS, IL-1β, and IL-6 in a dose-dependent manner), and the inhibition of astrocyte and microglial activation in an experimental stroke model in rodents [113]. Simvastatin also inhibited MMP-9 expression and BBB damage and reduced the risk of HT induced by tPA under experimental circumstances [114].

### 6.5. Others

Promising results have been published on the stroke-outcome-improving effects of antidiabetic drugs like rosiglitazone [115], free radical scavengers, e.g., edaravone [116], and high-density lipoprotein [117].

## 7. Discussion

This review had the objective of collecting the contributors of the hemorrhagic transformation of ischemic strokes mainly from a practical point of view, without aiming to provide an exhaustive list concerning the underlying molecular pathways.

The main components of the pathophysiological background were summarized as a starting point. First, the pathology of the ischemic phase of the damage was detailed, followed by the processes of the revascularization phase. Although the latter is crucial for tissue survival, and the possible treatment options like intravenous thrombolysis and mechanical thrombectomy aim to achieve it, revascularization also entails the potential for HT. The explanation of this at the cellular and tissue level is the triphasic permeability response during the restoration of blood flow. Based on a simple view of this issue, the initial hyperemia during the ischemic phase is caused by the acute disruption of the BBB due to energy and nutrient loss. The first and second phases of the following biphasic permeability increase are caused by oxidative stress, inflammation, extravasation of blood cells, and the processes aiming to restore BBB integrity and blood supply of the ischemic tissue, including angiogenesis. Reperfusion therapy adds to these complex risk factors of HT as recombinant tPA has been shown to disrupt BBB integrity via several ways, and MET may cause direct endothelial damage.

Among the clinical risk factors, high blood pressure is of pivotal importance. Treatment guidelines promptly define the blood pressure values required to be reached for revascularization methods, but there are no consensus guidelines for post-procedural blood pressure management. Actually, after reviewing the literature, it can be concluded that no strict values should be kept in the post-stroke period, but individualized management should be implemented. Patient characteristics and comorbidities need to be considered, and efforts should be made to be able to measure the autoregulatory capacity individually. Regarding hyperglycemia, it is noteworthy to mention that proper acute treatment, though required, does not help to reduce the risk of hemorrhagic transformation and a worse outcome. This fact reflects the importance of the optimal treatment of diabetes mellitus and other underlying, modifiable risk factors, which in turn requires patient partnership. Another interesting finding is that—just like in many other neurological and systemic disorders—the microbial composition of the gastrointestinal system may influence the stroke outcome. Keeping a healthy diet helps in the management of blood pressure, improves glucose metabolism, and promotes the development of a gut microbiota that is not harmful for the body. Further components of secondary stroke prevention strategies include lipid-lowering therapies, smoking cessation, and antithrombotic therapy [60]. Statins, being the mainstream of cholesterol-level-lowering therapy, not only reduce the progression of atherosclerotic plaques but also decrease BBB leakage and alleviate neuroinflammation at the site of ischemic injury [112,113,114]. Antithrombotic therapies represent a significant part of secondary stroke prevention. The use of thrombocyte aggregation inhibitors is recommended in the case of atherothrombotic stroke, while anticoagulants are recommended in the case of cardioembolic stroke [60]. Current available guidelines provide advice on their application in different clinical situations. Some main observations can be made as follows. In the case of TIA or minor stroke (NIHSS score < 3), when no thrombolysis or EVT is performed, dual-antiplatelet therapy with aspirin and clopidogrel is recommended for 21 days [20]. In more severe cases, monotherapy is suggested. If intravenous thrombolysis is performed with or without EVT, the antiplatelet therapy should be postponed 24 h, and the initiation should be preceded by a control cranial CT to exclude HT. If stenting is performed during or independent of EVT, dual-antiplatelet therapy should be started immediately and continued for three months [58]. Intensified antiplatelet therapies carry the risk of HT, even sICH, although the rate of mortality may remain unaffected [60]. In cases of cardioembolic stroke, anticoagulants have to be used, precisely warfarin in the case of valvular AF and the implantation of biological (for 3 months) or mechanical (lifelong) artificial valves [66]. In the case of nonvalvular AF, the use of direct thrombin and factor Xa inhibitors, such as apixaban and rivaroxaban, may have less risk of intracranial hemorrhage than patients on warfarin [65]. For the prophylaxis of DVT and PE in bedridden patients with severe ischemic stroke, LMWHs are recommended once a day. A combination of low-dose DOACs with an antiplatelet drug can be considered in patients with concomitant coronary or peripheral artery disease one month after their ischemic stroke or TIA [64].

As was mentioned in the introduction, the exact frequency of HT is difficult to determine because of the different definitions used in different studies. Another contributor to this is the potentially silent nature of this complication making it more common than would be first assumed. As Powers has concluded, HT may be present despite neurological improvement considering a patient’s symptoms or results of NIHSS scale ranging [67]. Clinicopathological observations showed that it is a considerable complication, even after 24 h of treatment, that is not or just partly attributable to rtPA’s effects as its half-life is fairly short, and the 24 h control cranial CT scan did not show HT. Therefore, a further control cranial CT a few days after the event or upon patient discharge, even without obvious clinical deterioration, should be considered for the future planning of secondary prevention and general management as well [118]. Another important finding of the previous clinicopathological studies is the importance of performing autopsies, as many HT cases remain undefined, the rate ranging between 35 and 53% and may be diagnosed only upon autopsy [119,120]. Certain biomarkers or a combination of them may be useful for the prediction of HT and promote a more precise clinical follow-up if a higher risk is suggested. The presence of the listed and detailed clinical risk factors (mainly hypertension), the mentioned CT signs (middle cerebral artery sign, signs of atherosclerosis, frank hypodensity on the initial CT scan that also affects the cortex) and the use of combined biomarkers altogether may help in the prediction of HT and in the selection of patients in the case of whom serial CT scans should be performed for optimal management. Also, in this context, the timing of antithrombotic treatment can be better planned.

## 8. Take Home Messages

The optimal timing of revascularization treatment for carefully selected patients and the individualized management of underlying diseases and comorbidities are important. Clinical practice encompasses comprehensive secondary stroke prevention measures including antithrombotic treatment, which can be simple in mild forms of the disease but may be highly challenging in severe, complicated cases. Valuable guidelines are available to help the work of clinicians; however, individualized management may be required when precise recommendations are missing. The measurement of biomarkers is time-requiring, and considering the narrow time window for the treatment of ischemic stroke, the baseline determination of biomarkers in the acute clinical setting is barely feasible and is also lacking therapeutic consequence. However, they may be useful for further patient management. It seems reasonable that instead of single biomarkers, a carefully chosen combination should be used to predict the chance of HT and clinical outcomes. Further molecular biological and proteomic studies are needed to find the most suitable biomarkers. Based on clinicopathological studies, HT of ischemic strokes is more common than previously thought; therefore, a more strict clinical follow-up, including serial cranial CT scans, can be suggested in selected cases.

## 9. Future Directions

A more harmonized definition of HT should be set for future studies. Regarding biomarkers, the uniformization of measurement methods should be reached from the laboratory part to make results comparable between different research and clinical sites. Another problem to be resolved is the determination of proper cutoff values again for comparable risk determination. Further randomized clinical studies are needed for both optimal risk factor management (like hypertension) and for the introduction of new auxiliary treatment options besides revascularization methods to avoid hemorrhagic complications. Based on our previous clinicopathological studies, it can be recommended to perform another cranial CT before the discharge of the patent, after the mandatory CT after 24 h after rtPA administration or EVT, actually, independent of both the treatment and the neurological progression, to evaluate HT.

## Figures and Tables

**Figure 1 ijms-24-14067-f001:**
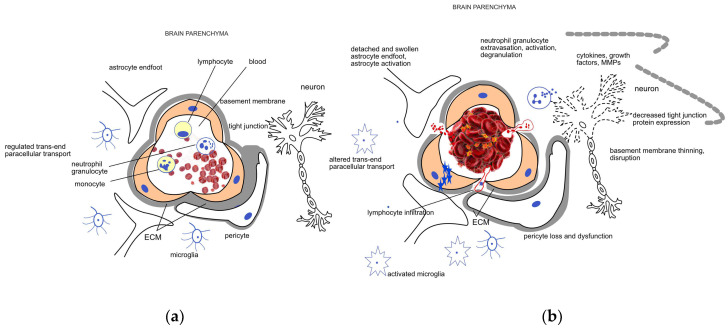
Schematic representation of the (**a**) healthy neurovascular unit (NVU) and blood–brain barrier (BBB) and (**b**) their disruption due to acute ischemic damage. The NVU is composed of endothelial cells, pericytes, the extracellular matrix (ECM) surrounding them, astrocytes, and neurons. The main components of the BBB are the endothelial cells, the different junctions between them, and the basement membrane. Endothelial cells, pericytes, and the surrounding ECM ensure a regulated trans- and paracellular transport required for the maintenance of the homeostasis in the brain parenchyma under healthy conditions. Mechanical occlusion of a vessel by a thrombus or an embolus, as shown on the right side of the figure, brings about a cascade of events including endothelial swelling, detachment and dysfunction of pericytes and astrocytes. The trans- and paracellular transport changes (represented by blue stars and red dots, respectively), leading to a sequel of pathophysiological processes including cytotoxic edema. Neutrophil cells are recruited to the damaged area; their extravasation, activation, and degranulation contribute to the neuroinflammatory processes mediated by the activated microglia and several soluble factors. Induction of proteases like matrix metalloproteases (MMPs) further enhance ECM degradation and disruption of the BBB.

**Table 1 ijms-24-14067-t001:** Phases of ischemic and reperfusional damage after acute ischemic stroke.

	Ischemic Phase	Revascularization Phase		
cerebral blood flow	occlusion, decreased cerebral blood flow	acute elevation of cerebral blood flow = hyperemia	decreased blood flow = hypoperfusion	
permeability	transcellular increase	initial reperfusional permeability	first phase of biphasic permeability	second phase of biphasic permeability
onset	immediately	upon spontaneous opening or thrombolysis or MET (minutes, hours)	3–8 (5) h	18–96 (72) h
cause	failed intracellular homeostasis, mitochondrial dysfunction	acute opening of the BBB, loss of cerebral autoregulation	cerebral metabolic depletion, microvascular obstruction, insufficient nutritional support	increased inflammatory activity, angiogenesis
underlying mechanisms	ATP depletion, excitotoxic glutamate efflux from neurons	disassembly of tight junctions	inflammatory and oxidative stress on the BBB, ECM degradation	imperfect tight junction reassembly, new assembly fails to reach the original paracellular impermeability
type of edema	cytotoxic		vasogenic	

**Table 2 ijms-24-14067-t002:** Clinical risk factors of HT and their potential resolution.

Pre-Stroke Risk Factors	Potential Resolution
• hypertension	individualized treatment
• hyperglycemia	individualized treatment
• advanced age	primary prevention
• liver fibrosis	primary prevention, supportive treatment
• altered iron homeostasis	use of neuroprotective substances
• antithrombotic treatment	individualized treatment
• low platelet count and mean platelet volume	individualized treatment
**Cranial CT, CTA Signs**	
• early ischemic sign	shortening time till reperfusion therapy, potent edema decrease, individualized treatment
• hyperdense middle cerebral artery sign	
• calcification of cerebral vessels	
• poor collateral circulation	
**Stroke Factors**	
• severe form of ischemic stroke	careful patient selection for reperfusion therapy
• reperfusion therapy	careful observation, optimal management

**Table 3 ijms-24-14067-t003:** Possible biomarkers predicting hemorrhagic transformation after ischemic stroke.

Biomarker	Abbreviation	Physiological Role	Predictive Value
matrix metalloproteinase 9	MMP9	proteolytic enzyme	sensitive and specific marker
cellular fibronectin	c-Fn	major component of the ECM	sensitive and specific marker
ferritin	NA	acute phase protein	neither specific nor sensitive enough
neutrophil-to-lymphocyte ratio	NLR	blood cells	low discriminative ability
S100 calcium-binding protein B	S100B	glial-specific protein	sensitive but not specific
soluble ICAM-1	sICAM-1	adhesive protein involved in inflammatory responses	sensitive but not specific
vascular adhesion protein-1	VAP-1	adhesion molecule	sensitive but not specific
semicarbazide-sensitive amine oxidase	SSAO	enzyme	

## Data Availability

Not applicable.

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
