# Peer review of "Hemorrhagic Transformation of Ischemic Strokes"

_ijms, 2023, doi:10.3390/ijms241814067_

Round 1
Reviewer 1 Report
This article addresses a critical aspect of ischemic stroke, a major global health concern, by delving into the potentially grave complication of hemorrhagic transformation (HT). Hemorrhagic transformation not only increases morbidity but also escalates the mortality rate, adding layers of complexity to a disease that is already a leading cause of death and disability worldwide.
One of the strengths of this summary is its thorough examination of the pathophysiology of hemorrhagic transformation. Understanding the underlying mechanisms is pivotal, as it not only sheds light on the disease process but also sets the stage for therapeutic innovations.
The manuscript holds promise with its valuable content but requires intensive revisions in terms of organization, language, and illustrations. Implementing these changes will significantly elevate the article's quality and readability, aligning it more closely with the expectations of a review piece.
Given that this is intended as a review article, there's an expected standard of organization and structure associated with such papers. While the manuscript does incorporate essential data, the overall presentation lacks coherence and can be challenging to navigate for readers.
The layout could benefit from a more distinct division of sections and better adherence to conventional review article structuring. The flow of information often feels sporadic, and readers might find it cumbersome to trace the logical progression of the narrative.
Several linguistic errors detract from the article's readability and professionalism. Notably:
Line 10: Capatalization in : Abstract: Background: ischemic stroke, (also in lines 13, 18)
Line 20: The word should be "avoid" instead of "avoide."
Line 42: The correct term is "searched" rather than "searced."
It's crucial that the entire document undergoes rigorous proofreading. For instance, the phrasing in line 82 can be more appropriately articulated as: "In subsequent sections of this article, we will delve into the aforementioned pathological processes, with an emphasis on their therapeutic options." Similarly, line 112 can be refined to: "Rather than adhering strictly to specific blood pressure (BP) values, clinicians should consider individual patient characteristics, such as the degree of reperfusion, infarct size, presence of carotid artery stenosis, antithrombotic therapy, and hemodynamic status. Personalized management should then be implemented based on these factors."
The freehand illustrations in Fig 1 a and b need improvement in terms of quality. The note "Own illustration." is extraneous and can be omitted. Moreover, the visuals should be aligned with the content discussed in the text. Table 1 does a reasonable job highlighting the risk factors of HT. However, the absence of a comprehensive illustration of Biomarkers is conspicuous and should be addressed.
While the manuscript sheds light on several aspects of hemorrhagic transformation (HT) in ischemic stroke, it misses out on addressing certain promising areas that have significant implications in clinical practice.
One conspicuous omission is the expanding role of antiplatelet therapy in the acute stroke treatment paradigm. On the one hand, intensification is linked with improved efficacy. On the other, this very increase in intensity is correlated with a heightened rate of HT, which can be detrimental. Such a nuanced topic demands a comprehensive discussion to provide clarity to clinicians on its risk-benefit ratio.
While the manuscript briefly touches upon the role of anticoagulation in cardioembolic strokes, it fails to delve into the intricacies of antiplatelet treatment for non-cardioembolic strokes. This omission is significant, given the pivotal role antiplatelet agents play in this context. A thorough discussion on this topic would greatly benefit the reader, particularly in understanding the decision-making process when it comes to selecting the appropriate therapeutic strategy for different types of ischemic strokes. (see PMID: 34063551)
Incorporating these additional sections will not only enhance the comprehensiveness of the review but will also provide clinicians with a more holistic understanding of the current state of treatment options and their associated challenges. The manuscript, in its current form, is rich in content, but these additions will undoubtedly make it more robust and beneficial for its intended audience.
Reviewer 2 Report
see below

minor editing of the English language required
Round 2
Reviewer 1 Report
Thank you for including me in the revision of the manuscript. All the earlier issues were tackled, and the MS improved considerably.